

# Effects of gait speed on paraspinal muscle activation: an sEMG analysis of the multifidus and erector spinae

Aleksandra Bryndal[1,2,*], Wojciech Nawos-Wysocki[1,*], Agnieszka Grochulska[1], Karol Łosiński[1] and Sebastian Glowinski[1,2]

[1] Institute of Health Sciences, Pomeranian University in Slupsk, Slupsk, Poland
[2] State Higher School of Vocational Education in Koszalin, Koszalin, Poland
[*] These authors contributed equally to this work.

Corresponding authors
Aleksandra Bryndal,
aleksandra.bryndal@upsl.edu.pl
Sebastian Glowinski,
sebastian.glowinski@upsl.edu.pl

## ABSTRACT

**Background.** The paraspinal muscles, such as the multifidus muscles and erector spinae muscles, play an important role in trunk stabilization and pelvic mobility during gait. Understanding how they are activated according to the speed of locomotion can facilitate the diagnosis and treatment of patients with these conditions. This study aimed to comprehensively analyze the activity of postural muscles (multifidus and erector spinae) using surface electromyography (sEMG) across a range of gait speeds.

**Methods.** The study group consisted of 31 students of physiotherapy at the Pomeranian University in Słupsk, including 20 women (64.51%) and 11 men (35.48%). The research process included an interview and participation in the sEMG survey. The sEMG examination of the musculus erector spinae (MES) and musculus multifidus (MM) was carried out in the supine, standing position, while walking on a treadmill, maintaining speeds: 1 km/h, 3 km/h, 5 km/h and 6 km/h, for 60 s at each stage and at maximum voluntary isometric contraction (MVIC). The percentage value of the amplitude of the sEMG recording in relation to maximum voluntary isometric contraction (%MVIC) was determined. The average rate of change in muscle activity was also assessed in relation to the respective changes in locomotion speed for MES and MM.

**Results.** The results show significant differences in the %MVIC parameter between sides (left *vs* right) for both muscle groups (MES and MM) in the supine position and in the standing position for the multifidus muscles. At the set speeds of 1 km/h, 3 km/h, 5 km/h and 6 km/h, the differences are not statistically significant. A decreasing trend in the rate of change in muscle activity was also observed for both muscle groups as gait speed increased. The most significant decrease is observed at average gait speeds of 3–5 km/h.

**Conclusions.** In the sEMG examination during rest, standing and walking, the activity of the right and left MES and MM muscles examined is not always equal. The activity of the muscles studied (MES and MM) increases with increasing gait speed. The rate of change in muscle activity decreases as gait speed increases.

# INTRODUCTION

Surface electromyography (sEMG) has established itself as a valuable, non-invasive tool in both clinical and research settings for assessing muscle and peripheral nervous system function (*Kotov-Smolenskiy et al., 2021*; *Hofste et al., 2020*). The versatility of sEMG extends to a wide range of applications, including rehabilitation diagnostics, movement testing, and biofeedback for robotics, prosthetics, and bioengineering.

Advancements in technology have significantly broadened the utility of sEMG. Integration with systems like Inertial Measurement Units (IMUs) and ongoing methodological refinements have positioned sEMG as a valuable asset in biomechanical movement analysis and practical rehabilitation (*Woodford & Price, 2007*; *Moreau et al., 2016*; *Glowinski, Blazejewski & Krzyzynski, 2017*). Furthermore, sEMG is proving invaluable for investigating the dynamic aspects of muscle function under natural movement conditions (*Glowinski et al., 2022*).

sEMG provides detailed insights into the coordination and activity patterns of muscles in the lower limbs, upper limbs, and trunk during various phases of gait. This information is crucial for diagnosing gait disorders, tailoring rehabilitation therapies, and designing effective prostheses and orthoses (*Ryu & Kim, 2014*; *Disselhorst-Klug, Schmitz-Rode & Rau, 2009*). Analyzing sEMG data allows for precise identification of muscle activation patterns, which are fundamental to diagnosis, treatment monitoring, and assessing rehabilitation progress (*Song, 2015*; *Karagiannopoulos et al., 2020*).

The insights provided by sEMG extend beyond clinical applications. By analyzing muscle activation patterns under varying conditions—such as changes in gait speed or execution of precise movements—researchers gain a deeper understanding of the central nervous system's role in controlling and coordinating movement (*Baudry, Minetto & Duchateau, 2016*; *Disselhorst-Klug, Schmitz-Rode & Rau, 2009*; *Balbinot et al., 2022*). This knowledge contributes significantly to our understanding of motor neurophysiology.

The sEMG, as opposed to the classical form of EMG, differs in the way it reads the potentials that arise and propagate within the muscle. This method uses special electrodes placed directly on the skin. The main principle of the electrodes that are used in sEMG is to read the muscle impulses that are underneath them. However, this brings with it a number of limitations and problems. These include the difficulty in correctly positioning the electrodes, reading the signal of the relevant muscle and preparing the epidermis. The patient's physiological factors, such as skin thickness, adipose tissue and their resistance, also have an important influence on the outcome of the test. Impedance is also an issue in relation to the transition itself between the epidermal phase and the electrode itself, where a poor connection can lead to disruption of received pulses or the creation of interference (*Disselhorst-Klug, Schmitz-Rode & Rau, 2009*; *Kotov-Smolenskiy et al., 2021*). In addition, the sEMG signal can be disturbed by electromagnetic noise generated by electronic devices or patient movements. Appropriate filtering and signal processing techniques are used to minimise the impact of these factors (*De Luca et al., 2010*).

While EMG provides valuable insights into muscle activation patterns, understanding the specific roles of individual muscles during complex movements like gait remains crucial

(*Chang et al., 2017*). The lumbar multifidus muscle plays a crucial role in maintaining segmental stability of the lumbar spine, as it connects individual lumbar vertebrae (*Lee et al., 2014*). During gait, the erector spinae and multifidus muscles exhibit characteristic activation patterns. The erector spinae demonstrates peak activity between 50–60% of the gait cycle, ceasing activation at toe-off (*Chang et al., 2017*). On the other hand, the multifidus displays its highest level of activation prior to the midpoint of the gait cycle, with the precise timing influenced by walking speed (*Crawford et al., 2016*). This suggests a complementary role for these muscles in maintaining spinal stability and facilitating efficient locomotion. Previous studies have noted that asymmetries in the activation of paraspinal muscles, such as the multifidus and erector spinae, are common during gait and may have implications for gait stability and efficiency (*e.g.*, *Saunders et al., 2005*; *Lee et al., 2014*). These differences may arise due to muscle imbalances, postural adaptations, or compensatory strategies, particularly in individuals with lower back pain or dysfunction (*Crawford et al., 2018*).

While the essential role of these muscles in gait stability is recognized, a comprehensive understanding of how their activation levels adjust to different walking speeds remains elusive. This research aims to address this gap by meticulously examining the relationship between various gait speeds and the sEMG activity recorded from the multifidus and erector spinae muscles. By illuminating how these muscles respond to changes in speed, our findings will offer valuable insights into the complex neuromuscular control strategies employed during locomotion. This knowledge has the potential to inform more targeted rehabilitation interventions for individuals experiencing gait-related impairments.

To achieve these objectives, this study aimed to comprehensively analyze the activity of postural muscles (multifidus and erector spinae) using surface electromyography (sEMG) across a range of gait speeds. Specifically, our objectives were to quantify the average amplitude of electrical potentials generated by these muscle groups during locomotion at various speeds and to examine how the relative contribution of activity within each muscle group shifts in response to changes in gait speed. We hypothesize that the activity of the erector spinae and multifidus muscles increases proportionally with gait speed, but decreases at higher speeds, reflecting adaptations in muscle coordination and recruitment patterns.

## MATERIALS & METHODS

### Participants

Forty people were initially enrolled in the study, of whom nine did not meet the inclusion criteria and were excluded from further analysis. The final study group consisted of 31 healthy subjects, students of physiotherapy at the Pomeranian University in Słupsk (20 women (64.51%) and 11 men (35.48%)). The mean age of the study participants was 22.5 years; (SD 2.4) years. The data were collected between 05.04.2023 and 22.06.2023. The purpose and protocol of the study were explained and each participant signed an informed consent form before participating. This study was approved by the Bioethics Committee at the District Medical Chambers in Gdańsk (KB-33/23).

## Selection criteria

Inclusion criteria for the study involved being over 18 years of age, being a physiotherapy student at the Pomeranian University in Słupsk, and signing an informed consent to participate in the study. Exclusion criteria for the study included individuals with a history of spinal injury or after spinal surgery, people with spinal and/or limb deformities, people with neurological diseases, pregnant women (due to static changes in the musculoskeletal system during pregnancy and possible resulting spinal pain), people who exceeded the safe maximum heart rate for the physical activity performed during the study.

## Research method

The study was based on the author's interview questionnaire and sEMG. These procedures were conducted at the Biomechanics Laboratory of the Department of Physiotherapy at the Pomeranian University in Słupsk. The duration of the examination of one person was approximately 20–25 min.

The interview questionnaire includes questions on anthropometric data such as age, gender, height (cm), weight (kg) and BMI (calculated as weight divided by the square of height) (kg/m$^2$). The interview questionnaire also includes questions relating to study inclusion and exclusion criteria.

The sEMG test was performed with an eMotion EMG device, equipped with 4 MT-WBA-1-EMG sensors and 12 Ambu Blue Sensor R (Ag/AgCl) electrodes measuring 57 × 48 mm. The sensors were connected wirelessly to the eMotion EMG receiver (Bluetooth module). The EMG signals were sampled at 1 kHz, with a common mode rejection ratio (CMRR) of 104 dB and a resolution of 14 bits. The sensor sensitivity was 1 µV/bit. We applied a band-pass filter between 20 and 450 Hz to reduce noise and isolate the relevant muscle activation frequencies. Electrodes were placed according to the recommendations of the European Project Surface EMG for Non-Invasive Assessment of Muscles (SENIAM) with amendments after (*Okubo et al., 2010*; *Soer et al., 2022*). The areas on the patient's body where the electrodes were placed were washed with soapy water and then disinfected with a 70% alcohol solution. If hair was present at the electrode sites, hair removal was performed. Once the electrodes were attached, the device was calibrated.

The location of the electrodes for the multifidus muscle (MM) was at the initial attachment near the L5/S1 vertebral transition (due to the most superficial location of the muscle in its course) at an electrode edge distance of 2 cm lateral to the spine, with a 2 cm gap between the electrode edges, in line with the course of the muscle fibres. The electrodes for the erector spinae muscles (MES) were placed 2 cm above the line defined by the most cranial (cephalic) vertices of the edge of the iliac crest, at a distance of the electrode edge 2 cm lateral to the spine, with a 2 cm gap between the electrode edges, following the course of the muscle fibres (Fig. 1).

## Test procedure

Patients underwent a single examination session consisting of three interconnected phases (Fig. 1). Firstly, participants were introduced to the course, indications and contraindications and the purpose of the study, and after giving their informed and written consent, they were interviewed.
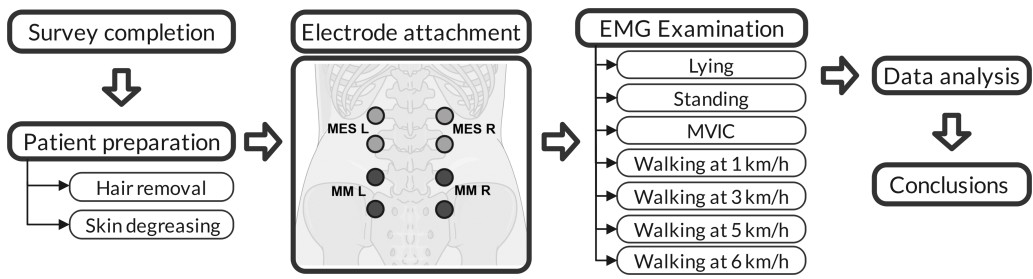

**Figure 1** **Experimental procedure flowchart.** Image source credit: Servier Medical Art: Bones by Servier, CC BY 4.0.

Secondly, participants were prepared for surface electromyography (sEMG). Further in this phase, initial sEMG measurements of the postural paraspinal muscles were taken in three positions: (1) at rest while lying prone with a bolster under the participant's abdomen to relax the examined structures; (2) while standing; and (3) during maximum voluntary isometric contraction (MVIC) (*Meldrum et al., 2007*; *Norasi, Koenig & Mirka, 2022*). When performing MVIC, the patient lay forward, with restraining straps around the shoulder girdle and shin, and performed a maximal torso extension. MVIC with simultaneous sEMG recording was performed to normalize the sEMG data. Each measurement lasted 60 s. Prior to commencing formal testing, all participants underwent a treadmill familiarization phase. During this phase, participants were allowed to walk on the treadmill at a self-selected comfortable pace for 5 min to acclimate themselves to the treadmill and the mechanics of walking.

In the third phase, sEMG was recorded during treadmill walking at speeds of 1 km/h; 3 km/h; 5 km/h and 6 km/h. The test time was 60 s for each speed. Muscle activity was measured only during gait. A 30-second rest was taken between testing at 3 km/h and 5 km/h and 5 km/h and 6 km/h. The subject stood on a stationary treadmill before each sEMG measurement. Before the sEMG measurement, the walking speed on the treadmill was increased until the target walking speed was reached. The sEMG measurement was performed only when the target speed was reached. The measurement lasted 60 s. After the 60-second measurement was completed, the treadmill speed was reduced to zero. After reaching zero, there was a 30-second break. Then, the procedures were repeated for each target walking speed. Heart rate was monitored at rest and during physical activity performed during the study using a medical finger pulse oximeter.

## Comparative methods

Following data acquisition during the test procedure, a series of comparative methods were employed to analyze and interpret the sEMG signals. These methods focused on quantifying muscle activity levels and their relationship to different gait speeds.

The data were converted to a percentage value of the amplitude of the sEMG recording in relation to maximum voluntary isometric contraction (%MVIC) according to the following relationship:

$$\%MVIC = y_{signal \forall v \rightarrow} \cdot y_{MVIC}^{-1} [\%] \tag{1}$$

where: %MVIC - current percentage value of MVIC,

$y\_(signal\ \forall \vec{v})$ - signal value of the examined muscle for each specified speed,

$y\_MVIC$ - signal value of the examined muscle during MVIC.

Due to the presence of outliers, the median was used to represent the relationship between the growth of the arguments—expressed as the rate of change in muscle activity—and the analysis of %MVIC changes across the given activity types and locomotion speeds. Lagrange polynomial interpolation from this relationship was then used to analyze the trend in more depth:

$$W_n(x) = \sum_{j=0}^{n} y_j \frac{(x-x_0)(x-x_n)}{(x_j-x_0)(x_j-x_n)} \Rightarrow W_n(x) = \sum_{j=0}^{n} y_j \prod_{k=1; k \neq j}^{n} \frac{x-x_k}{x_j-x_k} \tag{2}$$

$$W(x) = -\frac{y_{1\ km/h}(x-3)(x-5)(x-6)}{40} + \frac{y_{3\ km/h}(x-1)(x-5)(x-6)}{12}$$
$$-\frac{y_{5\ km/h}(x-1)(x-3)(x-6)}{8} + \frac{y_{6\ km/h}(x-1)(x-3)(x-5)}{15} \tag{3}$$

where: $y_{xkm/h}$—the median value of the received signal for a given muscle during the walking trial on a treadmill at a speed of $x$ km/h.

Equations (2) and (3) were used to determine the functions of the given muscles and to establish equivalent values. Finally, the relationships between argument increments for a given discharge, expressed as the rate of change in muscle activity, were also examined based on the established relationship:

$$\Delta_M = 1 + (y_{MVIC\%\forall(\vec{v}+n)} - y_{MVIC\%\forall\vec{v}}) \cdot y_{MVIC\%\forall\vec{v}}^{-1} \tag{4}$$

where: M—rate of change in muscle activity of the examined muscle;

$y_{MVIC\%\forall\vec{v}}$—percentage value of MVIC for each specified speed

$y_{MVIC\%\forall(\vec{v}+n)}$—percentage value of MVIC for each subsequent speed.

## Statistical analysis

Statistical analysis was conducted using IBM SPSS Statistics (version 29, IBM Corp., Armonk, NY, USA) and MedCalc Statistical Software (version 22.016, MedCalc Software Ltd, Ostend, Belgium). Graphical representations were generated using GraphPad Prism (version 10.1.0, GraphPad Software, Boston, USA). Quantitative variables are presented as the mean (standard deviation) [95% confidence interval], median, and range. Qualitative variables are summarized as frequencies and percentages. The normality of variable distributions was assessed using the Shapiro–Wilk, Anderson-Darling, and Kolmogorov–Smirnov tests with Lilliefors correction. Levene's test was used to evaluate the homogeneity of variances. For comparisons between two groups, an independent samples $t$-test was used when the assumptions of normality and homogeneity of variance were met; otherwise, the Mann–Whitney $U$ test was applied. Differences in the same variable across multiple groups, assuming a non-normal distribution, were assessed using the Friedman test, followed by *post-hoc* analysis if significant differences were found. A significance level of $p < 0.05$ was applied to all statistical tests.

**Table 1** Statistical description of the study group: Mean (SD); [95% CI]; Median; Range.

| Age (years) | Height (cm) | Weight (kg) | BMI |
|---|---|---|---|
| 22.5 (2.4) | 171.5 (8.8) | 70.4 (14.3) | 23.8 (3.8) |
| [21.6–23.4] | [168.2; 174.7] | [65.1; 75.6] | [22.4; 25.2] |
| 22 | 171 | 68 | 23.0 |
| 21–33 | 157–190 | 49–108 | 18.7–32.5 |

**Table 2** Percentage of sEMG recording amplitude with MVIC from selected muscle groups during lying, standing and set gait speeds ($N = 31$); Mean (SD), [95% CI], Median; Range.

| Muscle/Lead | Side | Lying | Standing | 1 km/h | 3 km/h | 5 km/h | 6 km/h |
|---|---|---|---|---|---|---|---|
| musculus erector spinae (MES) | Left (MES L) | 8.0 (4.5) [6.3; 9.7] 7.0 0.7–19.2 | 9.6 (5.3) [7.7; 11.6] 8.8 1.0–24.5 | 30.1 (60.2) [8.1; 52.2] 15.6 1.6–332.3 | 30.3 (48.5) [12.5; 48.1] 18.6 3.9–280.0 | 37.6 (41.1) [22.6; 52.7] 28.0 5.0–234.3 | 42.2 (35.1) [29.3; 55.1] 34.0 5.5–168.6 |
|  | Right (MES R) | 5.6 (4.8) [3.8; 7.4] 4.6 0.5–27.6 | 8.1 (6.6) [5.7; 10.5] 6.1 0.8–35.2 | 21.5 (31.8) [9.8; 33.2] 18.6 0.9–178.6 | 29.6 (32.0) [17.9; 41.3] 18.3 4.0–163.1 | 32.5 (28.0) [22.2; 42.8] 24.7 5.2–151.5 | 42.2 (34.4) [29.6; 54.9] 29.9 5.9–167.1 |
|  | p-value | 0.0059 | 0.0967 | 0.3904 | 0.9383 | 0.4429 | 0.8382 |
| musculus multifidus (MM) | Left (MM L) | 3.6 (2.4) [2.7; 4.5] 2.9 0.3–9.7 | 5.9 (4.2) [4.4; 7.5] 4.6 0.8–17.5 | 21.9 (15.5) [16.3; 27.6] 19.2 1.1–70.0 | 29.8 (22.7) [21.5; 38.1] 23.0 4.8–100.0 | 44.6 (33.4) [32.4; 56.9] 31.3 5.5–134.7 | 59.8 (43.9) [43.7; 75.8] 43.4 6.4–160.0 |
|  | Right (MM R) | 10.9 (5.9) [8.8; 13.1] 9.4 1.2–25.7 | 12.4 (6.2) [10.2; 14.7] 10.9 1.9–28.6 | 22.1 (11.7) [17.9; 26.4] 23.5 1.8–46.7 | 29.4 (16.5) [23.3; 25.4] 27.7 6.1–75.6 | 41.2 (30.0) [30.2; 52.2] 29.9 7.9–135.6 | 52.3 (40.1) [37.6; 67.0] 39.1 10.8–169.4 |
|  | p-value | 0.0001 | 0.0001 | 0.3713 | 0.5496 | 0.7675 | 0.5975 |

## RESULTS

Table 1 shows the basic characteristics of the study group. The average age of the respondents is 22.5 years. The majority of subjects had a normal (18.5–24.9) BMI (67.7%), overweight (25.0–29.9) was found in seven subjects, representing 22.5%, and obesity (>30) in three subjects (9.7%).

Table 2 shows the %MVIC for the discharge from the right and left musculus erector spinae (MES R and MES L) and the right and left musculus multifidus (MM R and MM L). It can be noted that a statistically significant difference was only shown when lying down for the musculus erector spinae left–right ($p = 0.0059$). Musculus multifidus left–right muscles obtained different values when lying down and standing ($p = 0.0001$). The results show significant differences between the sides (left *vs.* right) for both muscle groups (MES and MM) in the supine position and in the standing position for the multifidus muscles. In other cases, the differences are not statistically significant.

The results show significant differences between the sides (left *vs.* right) for both muscle groups (MES and MM) in the supine position and in the standing position for the multifidus muscles. In other cases, the differences are not statistically significant.

**Table 3** Functions for the leads (Lagrange polynomial interpolation).

| Lead | The resulting function |
| --- | --- |
| MES L | $-0.0744583x^3+1.4754x^2-3.43309x+17.5878$ |
| MES R | $0.06965x^3-3.21932x^2+9.29281x+12.7371$ |
| MM L | $0.420255x^3-3.21932x^2+9.29281x+12.7371$ |
| MM R | $0.597082x^3-5.63776x^2+16.9262x+11.5628$ |

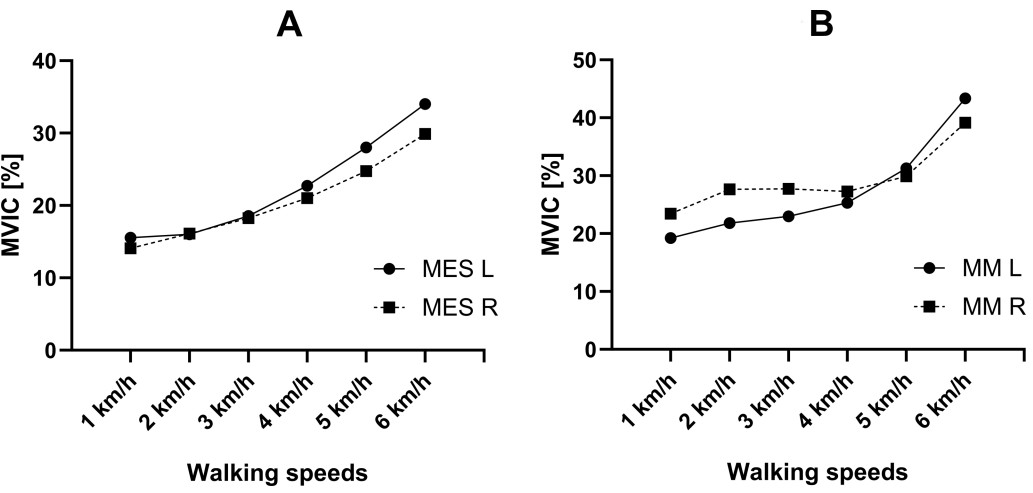

**Figure 2** Median percentage of MVIC values depending on locomotion speed, for erector spinae muscles (MES) (A) and multifidus muscles (MM) (B).

By converting discrete variables into continuous variables, it was possible to understand the evolution of changes in measurements in more detail (Table 3). Grade 3 polynomials allow the change in signal activity of individual muscles to be monitored.

On the basis of a multivariate function developed for a specific range of values (Table 3), an approximation of the locomotion speed arguments that were not directly the object of measurement was performed (Fig. 2).

The greatest differences for the MES L and MES R muscle occur at 5–6 km/h (12.4–12.9%). The smallest differences were observed for speeds of 2–3 km/h (0.4−1.6%). For the MM L and MM R muscles, the largest differences were in the range of gait speeds of 1–3 km/h (18.7–23.6%), while the smallest differences were for speeds of 4–5 km/h (4.5−7.5%). From the data obtained, it is possible to estimate the common points of the approximation curves, *i.e.,* to indicate the gait velocity at which the muscles concerned show the same activity, as read by the sEMG.

**Table 4  Comparative table of identical %MVIC participation values for selected muscles for the designated domain v1.00; 6.00.**

| Muscle 1 | Muscle 2 | v (km/h) | Shared %MVIC value |
|---|---|---|---|
| MES L | MES R | 1.87462 | 15.8464 |
| MES L | MES R | 2.64026 | 17.4381 |
| MM L | MM R | 4.59314 | 28.2257 |
| Sum of MES L & MM L | Sum of MES R & MM R | 2.97785 | 41.4412 |

**Table 5  Friedman test for individual discharges.**

| Parameters for the Friedman test | MES L | MES R | MM L | MM R |
|---|---|---|---|---|
| $X^2$ | 131,804 | 129,898 | 149,851 | 137,481 |
| Df | 5 | 5 | 5 | 5 |
| $p$ | 0.000 | 0.000 | 0.000 | 0.000 |

**Notes.**

Table description: $X^2$, Chi-square test result; Df, degrees of freedom; $p$, asymptotic significance.

Table 4 presents an analysis of the correlations between the functions describing muscle activity during locomotion, as represented by the sEMG signal.

The results are presented for the individual leads in Table 5. The results presented are statistically significant ($p < 0.05$) indicating that there are differences between the measurements compared in terms of the severity of the dependent variables concerned.

No statistically significant difference was observed between the left and right sides (Table 6). Figure 3 shows the mean value of the rate of change in muscle activity for the respective changes in locomotion speed for the musculus erector spinae (MES) (Fig. 3A) and the multifidus muscles (MM) (Fig. 3B). In Fig. 3A, we can see the changes in rate of change in muscle activity values for the musculus erector spinae (MES) on the left (MES L) and right (MES R) as a function of gait speed. The rate of change in muscle activity values for MES L and MES R decrease with increasing speed, with the most significant decrease observed for values of 3–5 km/h (MES R). Figure 3B shows the changes in rate of change in muscle activity values for the multisided muscles (MM) on the left (MM L) and on the right (MM R). As with the musculus erector spinae (MES), the rate of change in muscle activity value for the multifidus muscles (MM) decreases with increasing gait speed. The biggest drop is seen at speeds of 3–5 km/h, especially for MM R.

## DISCUSSION

This study investigated the relationship between muscle activity, measured by surface electromyography (sEMG) average amplitude values, and gait function. sEMG is a widely utilized diagnostic tool in both clinical and research settings (*Baudry, Minetto & Duchateau, 2016*; *Merletti & Muceli, 2019*; *Falla, 2016*).

**Table 6 Median growth factor results from sEMG measurements ($N = 31$): GM (SD); 95%CI$_{GM}$; Me, Min-Max.**

| Muscle | Left | Right | $p$-value |
|---|---|---|---|
| | Growth factor $v = 1 \rightarrow 3$ km/h | | |
| Musculus erector spinae (MES) | 1.25 (1.29) | 1.51 (2.08) | 0.1856[a] |
| | 1.08–1.45 | 1.22–1.87 | |
| | 1.27 | 1.31 | |
| | 0.64–8.29 | 0.64–11.60 | |
| Musculus multifidus (MM) | 1.40 (2.03) | 1.37 (1.62) | 0.9607[a] |
| | 1.16–1.68 | 1.17–1.59 | |
| | 1.26 | 1.27 | |
| | 0.68–12.23 | 0.96–10.20 | |
| | Growth factor $v = 3 \rightarrow 5$ km/h | | |
| Musculus erector spinae (MES) | 1.40 (0.47) | 1.23 (0.77) | 0.1039[a] |
| | 1.26–1.55 | 1.05–1.45 | |
| | 1.36 | 1.29 | |
| | 0.67–3.47 | 0.31–5.11 | |
| Musculus multifidus (MM) | 1.61 (1.03) | 1.30 (0.40) | 0.5035[a] |
| | 1.24–1.99 | 1.18–1.44 | |
| | 1.28 | 1.27 | |
| | 0.90–5.84 | 0.80–2.72 | |
| | Growth factor $v = 5 \rightarrow 6$ km/h | | |
| Musculus erector spinae (MES) | 1.17 (0.21) | 1.26 (0.40) | 0.2077[a] |
| | 1.08–1.26 | 1.13–1.40 | |
| | 1.23 | 1.26 | |
| | 0.62–1.56 | 0.53–2.90 | |
| Musculus multifidus (MM) | 1.30 (0.35) | 1.23 (0.32) | 0.1928[a] |
| | 1.18–1.43 | 1.12–1.34 | |
| | 1.25 | 1.19 | |
| | 0.53–2.37 | 0.71–2.06 | |

**Notes.**
[a] Mann–Whitney $U$ test.

Our findings demonstrate a significant increase in muscle activity across the studied muscles as gait speed increases. This observation aligns with previous research demonstrating an elevation in the %MVIC contribution of paraspinal muscles during faster gait speeds. *Crawford et al. (2018)* also observed that peak muscle activity was localized in the lower multifidus and lumbar erector spinae, particularly in young adults walking at 4 km/h on a 10% incline. Supporting our results, *Saunders et al. (2005)* and *Lee et al. (2014)* similarly reported increased multifidus and erector spinae muscle activity with escalating gait speeds. Specifically, they noted heightened activity in these muscles when locomotion speed increased from 3 to 6 km/h, using electrode placements at L3 for the erector spinae and L5 for the multifidus (*Saunders et al., 2005*; *Lee et al., 2014*). Furthermore, *Crouch et al. (2018)* suggest that sEMG may provide enhanced data quality and efficiency when used in real-time motion studies, such as the present investigation.

This study demonstrated that walking exercises activate the erector spinae muscle (MES) and multifidus muscle (MM) in the lower back region. Furthermore, high-speed walking exercises activate MES and MM to a greater extent than low-speed walking exercises. Previous studies have observed unilateral atrophy of the multifidus muscle

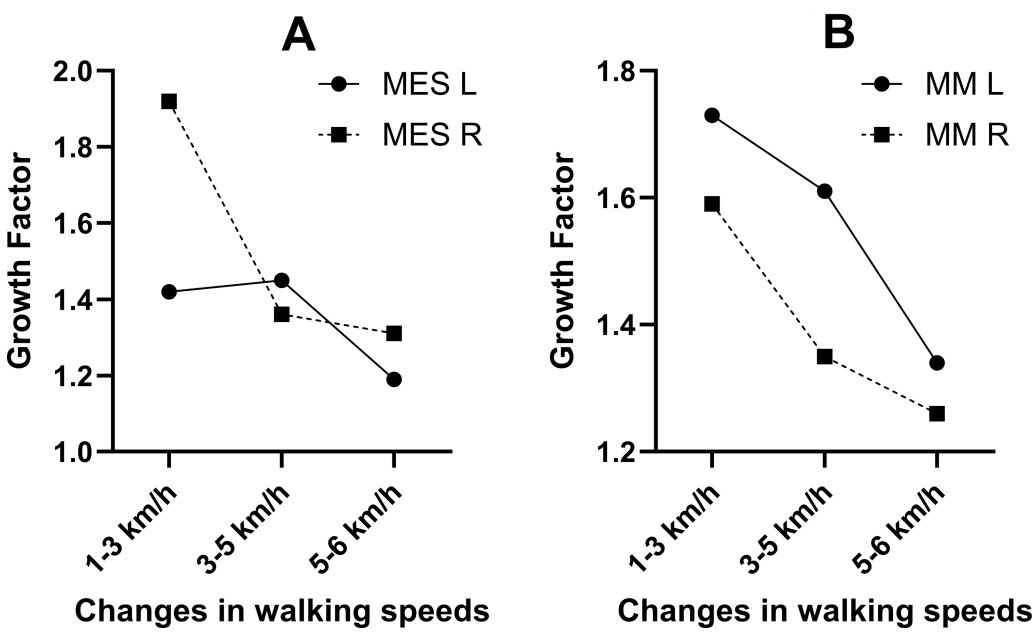

**Figure 3** Average growth factor value for respective locomotion speed changes for erector spinae muscles (MES) (A) and multifidus muscles (MM) (B).

on the symptomatic side and level in patients with low back pain (LBP), as well as a positive correlation between the duration of LBP symptoms and the degree of multifidus muscle atrophy (*Barker, Shamley & Jackson, 2004*; *Hides et al., 2008*; *Kjaer et al., 2007*). Our findings indicated that walking exercises activate the MES and MM in the lumbar region. Based on our results and those of other studies (*Ghamkhar & Kahlaee, 2015*), we suggest that walking exercises, particularly high-speed walking, may be beneficial for strengthening the lumbar muscles. Walking exercises can be easily implemented in physiotherapy due to their accessibility and appeal. Additionally, high-speed walking has been shown to improve cardiovascular and respiratory endurance.

Our study found significant differences in muscle activity between the left and right sides for both the multifidus and erector spinae muscles, particularly in the supine and standing positions. These findings align with previous research, which has suggested that such asymmetries can affect gait stability and efficiency. For example, *Saunders et al. (2005)* observed similar left–right differences in muscle activity and noted that such imbalances could lead to compensatory movement patterns, potentially contributing to gait disorders or musculoskeletal pain. Further research is needed to explore how these asymmetries affect functional gait performance and whether interventions aimed at reducing these imbalances could improve overall gait efficiency.

Similar to *Anders et al. (2007)*, our study showed higher values for multifidus muscles at 4 km/h, 5 km/h, and 6 km/h compared to the erector spinae muscle, suggesting increased demand for segmental stabilization during faster, more demanding walking. Interestingly, *Weber et al. (2017)* tested spinal and lower limb muscle activity in nine young men under

natural and more challenging walking conditions and reported superficial multifidus as the only tested muscle whose activity differed across walking conditions, with higher activity and longer duration occurring during demanding walking. It is difficult to speculate whether this finding has mechanical or metabolic underpinnings related to walking, considering our small sample size and limited age range; however, increasing muscle activation with changes in walking speed has implications for rehabilitation regarding exercise tolerance. Further investigation involving larger samples across wider age groups to elucidate potential age-related changes in spinal muscle activation would be beneficial.

This paper presents an innovative approach to the study of paraspinal muscle activity. An interdisciplinary approach combining medicine and technology, as well as mathematical modelling enables a broader view of the study of muscle activity. Using the proposed methodology, it is possible to continuously monitor the activity of each muscle analysed on both sides of the body. This contributes to a better understanding of the mechanics of the multifidus muscles and erector spinae muscles as a function of gait. Analysis of the interdependencies between the functions describing muscle activity during locomotion, represented by the sEMG signal, opens the way to identifying common points for these functions. These points represent the specific movement speeds at which identical sEMG signal amplitude is observed for the muscles studied, despite the potentially different biomechanical roles played by these muscles in the locomotion process (Table 4). This interdisciplinary approach, combining biomechanics with sEMG signal analysis, offers the possibility of accurately determining the speed of locomotion, at which the muscles involved in the movement show a similar intensity of activation. This allows the identification of specific transition points, where the dynamics of the movement change and the pattern of muscle activation adapts to the new conditions. In addition, this makes it possible to establish velocity ranges in which a clear change in the pattern of muscle activation is observed. Analysis of these changes may indicate a shift to a different movement strategy, such as a smooth change of speed or stabilization, or the involvement of additional muscle groups to ensure stability and movement efficiency.

In the study presented here, the percentage value of the amplitude of the sEMG recording in relation to maximum voluntary isometric contraction (%MVIC) at a given gait speed was determined for each muscle tested. These values varied from muscle to muscle, highlighting the specificity of the response of individual muscle groups to this type of activity. The use of interpolation allowed discrete data to be transformed into a continuous function, which has important practical implications. In addition to confirming the possibility of estimating muscle activity from a few measurements, the %MVIC values derived in the study provide a preliminary picture of the formation of muscle activity during gait. This model allows the activity of any muscle tested to be analysed at any speed within the test interval, significantly increasing the usefulness and flexibility of the results presented.

This study provides new insights into the activation patterns of the erector spinae (MES) and multifidus (MM) muscles during gait at varying speeds. Our findings indicate that the activity of these muscles increases with gait speed, with significant differences in activation observed between the left and right sides in certain positions. Moreover, a notable decrease in the rate of change in muscle activity was identified at moderate gait speeds (3–5 km/h).

These findings suggest that the central nervous system adjusts the recruitment patterns of these muscles as gait speed increases, which has not been sufficiently detailed in prior studies.

The main concern expressed by most researchers in the context of conducting muscle activity studies using surface EMG is the uncertainty of the results obtained and the potential interference of signals (noise) from different sources, which can lead to distortion of the data received. A number of comparative studies have been conducted to assess the effectiveness of two EMG techniques (*Crouch et al., 2018*), also in the context of the lumbar spine, where the action of the multifidus and erector spinae muscles was also investigated, so in developing the methodology for this study, the findings and reports of many authors were used to minimise the risk of repetition of their errors and to reduce potential limitations due to inaccurate preparation or conduct of the study (*Stokes, Henry & Single, 2003*; *Saunders et al., 2005*; *Okubo et al., 2010*; *Hofste et al., 2020*).

It is worth noting some further factors that may have had some influence on the results of the observations. Limited access to a study group that was age-homogeneous and highly selected may influence the observed symptoms and results, which are not necessarily representative of the general population. In addition, narrowing the study group to only physiotherapy students may result in limited readings due to a lifestyle that primarily combines study and work, and an awareness of the need for physical activity in the prevention of lower back pain. The presence of adipose tissue may have interfered with the normal flow of current, cutting off the limits, which is a limitation especially among overweight or obese people. Additional systems, such as the inertial measurement unit (IMU), should be used in future studies. It would also be interesting to compare the results of tests using the two systems: the sEMG and the needle system, which extracts activity information directly from the targeted muscle.

## CONCLUSIONS

This study investigated the activity of the multifidus (MM) and erector spinae (MES) muscles during gait at varying speeds using surface electromyography (sEMG). Our findings demonstrate that overall muscle activation within both groups increases as gait speed accelerates. Conversely, the rate of change in muscle activation exhibits a decreasing trend with increasing gait speed. Notably, sEMG examination revealed asymmetries in muscle activity between the right and left MES and MM muscles during rest, standing, and walking. These results highlight the dynamic interplay between gait speed and postural muscle activation, contributing to a better understanding of neuromuscular control during locomotion. These findings provide insights into an optimal walking exercise protocol for strengthening lumbar paraspinal muscles.

### Funding

The authors received no funding for this work.

### Competing Interests

The authors declare there are no competing interests.

### Author Contributions

- Aleksandra Bryndal conceived and designed the experiments, performed the experiments, analyzed the data, prepared figures and/or tables, authored or reviewed drafts of the article, and approved the final draft.
- Wojciech Nawos-Wysocki conceived and designed the experiments, performed the experiments, analyzed the data, prepared figures and/or tables, authored or reviewed drafts of the article, and approved the final draft.
- Agnieszka Grochulska analyzed the data, authored or reviewed drafts of the article, and approved the final draft.
- Karol Łosiński conceived and designed the experiments, analyzed the data, authored or reviewed drafts of the article, and approved the final draft.
- Sebastian Glowinski performed the experiments, analyzed the data, prepared figures and/or tables, authored or reviewed drafts of the article, and approved the final draft.

### Human Ethics

The following information was supplied relating to ethical approvals (i.e., approving body and any reference numbers):

The Bioethics Committee at the District Medical Chambers in Gdańsk (KB-33/23).

### Data Availability

The raw measurements are available in the Supplementary File.

### Supplemental Information

Supplemental information for this article can be found online at http://dx.doi.org/10.7717/peerj.19244#supplemental-information.

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
