# Peer review of "Effects of gait speed on paraspinal muscle activation: an sEMG analysis of the multifidus and erector spinae"

_PeerJ, doi:10.7717/peerj.19244_

## Round 0.1 · original submission · Major Revisions

The reviewers have gone to great lengths to provide detailed feedback, highlighting several points in the experimental design and data analysis, which need addressing. In particular, clarification is needed on the main aim of the article, in analyzing motor units, where this is currently lacking and the discordance between the finding of the paper and the analyses which have been undertaken.

·

Basic reporting

Thank you for giving me the opportunity to review this manuscript. The primary aim of the study was to analyze the activity of motor units in selected postural muscles assessed by sEMG in relation to gait speed. Despite the stated main objective focusing on the analysis of motor units, it appears that no such analysis has been conducted, which is a significant oversight. Additionally, the manuscript lacks any reference to prior research or discussion on motor unit activity during walking, which is crucial for contextualizing the results and demonstrating the study's relevance. The absence of a clearly stated hypothesis further complicates the understanding of the study's goals and conclusions. It is also concerning that analytical methods, which should have been detailed in the Methods section, are instead included in the Results section. Moreover, the title of the paper is ambiguous and does not effectively reflect the study's content, making it difficult for readers to predict the focus of the research from the title alone.

Experimental design

Although this overlaps with basic reporting, the study appears to primarily aim at analyzing motor units; however, no such analysis has been conducted. As a result, it seems that the research methods employed are not adequate to achieve the stated objectives. While there are methods like the deconvolution method or clustering method to estimate motor unit activity from sEMG, these approaches have not been referenced in the manuscript. The details regarding sensor locations during surface electromyography measurement are well described, yet the manuscript lacks sufficient explanation of the sampling methods and post-processing techniques.

Validity of the findings

The main finding of this study seems to suggest that the activity of the motor units in the right and left MES and MM muscles is not always identical. However, it appears that the motor unit activity has not been verified in this study. This discrepancy between the purported findings and the actual analyses performed raises concerns about the validity of the conclusions drawn.

Additional comments

Specific Comments
• The necessity of using Lagrange polynomial interpolation was unclear. Please explain in detail for a physiologists.
• The analysis interval is not specified.
• It is recommended to analyze and discuss the data for each walking phase separately.
• It is recommended to consider discussing the potential adverse effects of the observed differences between the left and right sides on walking. Understanding these implications can be critical for interpreting the functional significance of the findings.

Reviewer 2 ·

Basic reporting

This study examined the muscle activation patterns during gait in young, healthy adults (n=31). It offers an interesting contribution aimed at enhancing understanding of postural muscle activations (specifically the lumbar multifidus and erector spinae) at various treadmill speeds. However, I have a few comments regarding the manuscript.

The introduction is relevant and effectively conveys the main objective of this research. However, certain concepts require further detail particularly regarding the multifidus and erector spinae muscles which are not mentioned. Introducing the preferred activation phases of these muscles during gait in healthy individuals would be especially valuable.

I did not find titles or descriptions for the figures. Furthermore, I recommend changing the units of speed in the manuscript as well as in Figures 2 and 3. Specifically, the use of “km/h” should be replaced with “m/s” to align with international standards.

Finally, the structure of this paper and the raw data appear to conform to PeerJ's standards. The writing quality of this document makes it easy to understand, which is a strength. However, some adjustments to syntax and clarity in a few sentences would further enhance the overall quality of the manuscript.

Title:
Some terms could be adjusted for consistency, such as changing “analysing” to “analyzing”.

Introduction:
Line 57: The phrase “muscle activity recording method” could be enhanced by adding a determiner before “muscle activity,” such as “the” or “a”, for clarity

Line 85-86: The sentence, “This uses special electrodes placed directly on the skin…”. Could be clarified by adding “This method uses…” for a better clarity.

Materials and Methods:
Line 111: To improve clarity, consider revising “The research was collected between 05.04.2023 and 22.06.2023.” to “The data were collected…” as this phrasing more precisely reflects the collection of data over the specified period.

Line 111-112: Consider replacing “was explained” with “were explained” to ensure subject-verb agreement.

Line 119: Consider replacing “involved” by “included” to improve readability.

Line 119-120: To avoid redundancy, consider replacing “people with a history of spinal injury or after spinal surgery” with “individuals with a history of spinal injury or previous spinal surgery”.

Line 126: For clarity, I suggest “These procedures were conducted” instead of “They were conducted…”.
In lines 151, 154, and 155, the term “stage” is unclear. Do the authors mean that participants attended multiple sessions, or is “stage” being used to indicate steps in a process (e.g., firstly, secondly etc.)?

Line 156: To avoid confusion, consider replacing “lying forward” by “lying prone” for clarity.

Line 160: Consider adding “a” before “maximal torso extension” for grammatical accuracy.

Line 161: In scientific writing, the American English spelling “normalize” is generally preferred over “normalise” for consistency.

Results:
Line 196: Remove the % symbol after 100.

Discussion :
Line 292: Consider replacing “with sEMG” by “using sEMG”

Line 296: The term “recognize” is generally preferred over “recognise” for consistency in scientific writing.

Line 300: Consider replacing “increase the activity of the paraspinal muscles in the %MVIC contribution…” with “increase the %MVIC contribution of the paraspinal muscles…” for clarity.

Line 328: Consider replacing the term “stabilisation” with “stabilization” to follow American English spelling conventions.

Line 369: To improve readability, consider changing “…from the muscle in question” to “from the targeted muscle”.

Conclusion:
Line 380: I suggest the term “miniaturisation” instead of “miniaturization”.

Experimental design

This research aligns well with the scope of the journal. However, I believe the Materials and Methods section requires additional precision. I will address the following points in the order in which they appear in the Methods section.

Participant profile)
- The authors did not specify whether the study population was athletic or sedentary, which could directly impact the level of voluntary activation measured via EMG.
- In the Methods section, the authors state on lines 122-123: 'people who exceeded the safe maximum heart rate for the physical activity performed during the study.' However, no equipment for measuring heart rate is listed among the materials used.

sEMG)
- Could you clarify why you chose surface EMG (sEMG) instead of intramuscular EMG (iEMG) for electromyographic analysis of these deep muscles? A study by Hofste et al. [1] also highlights greater variability in sEMG measurements compared to iEMG, as well as the difficulty of isolating the lumbar multifidus signal with sEMG.
- The EMG sampling rate, data processing methods used, and whether filtering was applied are not specified in the manuscript.

% MVIC)
- Did each participant maintain an MVIC for 60 seconds, or is this a typo on line 161? Holding an MVIC for 60 seconds seems unfeasible, as this task is intended to represent a maximal effort.

Walking trials)
- Could you clarify whether participants underwent a habituation phase on the treadmill before performing the experimental conditions?
- Could you clarify whether EMG acquisition began upon reaching the target speed or during the treadmill's acceleration phase? Additionally, did the experimenters wait for the gait pattern to stabilize before starting EMG acquisition?

[1] Hofste, A., Soer, R., Salomons, E., Peuscher, J., Wolff, A., van der Hoeven, H., Oosterveld, F., Groen, G., & Hermens, H. (2020). Intramuscular EMG Versus Surface EMG of Lumbar Multifidus and Erector Spinae in Healthy Participants. Spine, 45(20), E1319–E1325.

Validity of the findings

Thank you for providing the raw data. Please could you explain why, at rest in the lying position, the average MVIC activation percentage is 8% on average for the left erector spinae muscle group and 10.9% on average for the right multifidus muscle?

Additional comments

This is an interesting article, but it would benefit from additional details in the Methodology section to improve clarity and enhance reproducibility. Additionally, some rephrasing may be needed to ensure an international audience can fully understand the text. I suggest having a colleague proficient in English and familiar with the subject matter review your manuscript or, alternatively, consider using a professional editing service

·

Basic reporting

Your introduction would benefit from additional detail. I recommend enhancing the description between lines 64 and 73 to offer stronger justification for your study, particularly by elaborating on the specific knowledge gap you aim to address. Additionally, lines 66 and 71-73 present repetitive ideas; consider consolidating these to improve clarity and coherence.
The English language should be improved to ensure that an international audience can clearly understand your text. Some examples where the language could be improved include lines 290–309 the current phrasing makes comprehension difficult. I suggest you have a colleague who is proficient in English and familiar with the subject matter review your manuscript, or contact a professional editing service.

Experimental design

Have you considered eliminating the deceleration and acceleration phases at the beginning and end of walking in your methodology? This could impact your analysis by providing data on the dynamics of movement and its variations.

In the initial methodology, it would be better to test the physical activity level of your population. Have you considered using a test like the Ricci-Gagnat test, for example?

In the conclusion, there’s a tendency to discuss a comparison of the techniques used to measure muscle activity, which doesn’t align with the initial objective.

Validity of the findings

In the conclusion, there’s a tendency to discuss a comparison of the techniques used to measure muscle activity, which doesn’t align with the initial objective.

How do you explain this result, which is a decreasing trend in growth factor values for both muscle groups observed as walking speed increased, with the most significant changes noted at moderate speeds (3-5 km/h)?

---

## Round 0.2 · Major Revisions

The reviewers have provided a comprehensive evaluation of the manuscript and highlighted numerous areas for clarification. Accordingly, the manuscript is in need of major revisions.

·

Basic reporting

Thank you for your response. While there have been improvements in the description of the objectives, significant doubts still remain regarding the literature review and the explanation of the study's novelty.

Previous Comments and Author Responses:
Point 2 Q2: Additionally, the manuscript lacks any reference to prior research or discussion on motor unit activity during walking, which is crucial for contextualizing the results and demonstrating the study's relevance.
Point 2: We thank the reviewer for this valuable suggestion. We have rewritten the introduction and objectives to make them more readable and to show the purpose of our study. We have presented the objectives of the work more precisely, which will show the exact intention of our study. We have included information on the activation of the multifidus and erector spinae muscles during walking. In the discussion in the second paragraph, we have discussed gait and its change depending on walking speed.

New Comments for Point 2:
I understand that the objective is not to elucidate the characteristics of motor units but to clarify how the activity of the multifidus and erector spinae muscles changes depending on walking speed. However, there remain doubts about the background explanation and the study's novelty. Numerous studies have been conducted on the activity of the multifidus and erector spinae muscles depending on walking speed (see the following reference list), but only two are introduced in the Background section of this paper, indicating a lack of literature review on previous studies. While it is not necessary for the authors to cite all of them, the current explanation makes it very unclear how much is already known.
Furthermore, regarding how the activity of the multifidus and erector spinae muscles changes with walking speed—which is the objective of this study—it appears that many studies have already clarified this (Crawford et al., 2016; 2018; Anders et al., 2007; Lee et al., 2014; Ghamkhar & Kahlaee, 2015; Zoffoli et al., 2016; Butowicz et al., 2018), raising significant doubts about the novelty of this research.

Reference List
Crawford, R., Gizzi, L., Dieterich, A., Ni Mhuiris, Á., & Falla, D. (2018). Age-related changes in trunk muscle activity and spinal and lower limb kinematics during gait. PloS One, 13(11), e0206514. https://doi.org/10.1371/journal.pone.0206514
Anders, C., Wagner, H., Puta, C., Grassme, R., Petrovitch, A., & Scholle, H. C. (2007). Trunk muscle activation patterns during walking at different speeds. Journal of Electromyography and Kinesiology, 17(2), 245–252. https://doi.org/10.1016/j.jelekin.2006.01.002
Lee, H. S., Shim, J. S., Lee, S. T., Kim, M., & Ryu, J. S. (2014). Facilitating effects of fast and slope walking on paraspinal muscles. Annals of Rehabilitation Medicine, 38(4), 514–522. https://doi.org/10.5535/arm.2014.38.4.514
Ghamkhar, L., & Kahlaee, A. H. (2015). Trunk muscles activation pattern during walking in subjects with and without chronic low back pain: a systematic review. PM&R, 7(5), 519–526. https://doi.org/10.1016/j.pmrj.2015.01.013
Zoffoli, L., Lucertini, F., Federici, A., & Ditroilo, M. (2016). Trunk muscles activation during pole walking vs. walking performed at different speeds and grades. Gait & Posture, 46, 57–62. https://doi.org/10.1016/j.gaitpost.2016.02.015
Butowicz, C. M., Acasio, J. C., Dearth, C. L., & Hendershot, B. D. (2018). Trunk muscle activation patterns during walking among persons with lower limb loss: Influences of walking speed. Journal of Electromyography and Kinesiology, 40, 48–55. https://doi.org/10.1016/j.jelekin.2018.03.006

Previous Comment and Response:
Point 3 Q3: The absence of a clearly stated hypothesis further complicates the understanding of the study's goals and conclusions.
Point 3: We thank the reviewer for this valuable suggestion. We have revised the objectives to make the purpose of our work more explicit.

New Comment for Point 3:
While the objectives have become slightly more detailed, the hypothesis is still not stated.

Experimental design

An acceptable response has been provided.

Validity of the findings

Because there was no description of the findings in the first paragraph of the Discussion, the new findings of this study are unclear. In the second paragraph in the discussion, the findings of previous studies are compared with those of this study, and I understand that the results are consistent with previous research. However, it is not clear what new insights have been added beyond the existing studies.

Additional comments

Previous Comment and Response:
Point 12 Q12: It is recommended to consider discussing the potential adverse effects of the observed differences between the left and right sides on walking. Understanding these implications can be critical for interpreting the functional significance of the findings.
Point 12: Thank you for this comment. We have changed the approach in the introduction where we have removed information on the change in activity of the multifidus and erector spinae during gait. We have also addressed this topic in the discussion. In future studies, we will take into account these valuable comments and extend our research with the suggestions provided.
New Comment for Point 12:
In both the Introduction and Discussion, I did not find any introduction or discussion of previous studies regarding left-right differences in the activity of the multifidus and erector spinae muscles during walking.

Minor Comments
• Abstract: The revisions made in the main text are not reflected.
• Reference List: There appears to be an issue with Gekht (1990):
Gekht B. 1990. Teoreticheskaya i klinicheskaya elektromiografiya [Theoretical and clinical electromyography]. Lvov, 1990, 229 p(rus).

Reviewer 2 ·

Basic reporting

I thank the authors for this comprehensive new version, which provides a clearer understanding of their research focus. I have some suggestions:

At line 250 in the revised version, the word "conducted" appears twice in succession.

Please note, the font size is not the same, particularly between lines 278 and 292.

At line 292 in the revised version, the authors can limit the redundancy of “walking” by simplifying the sentence. For instance: "sEMG was recorded during treadmill walking at speeds...".

I suggest changing ‘for every the specified speed’ at line 358 and ‘for every the subsequent speed’ at line 359 to ‘for each specified speed’ and ‘for each subsequent speed’ respectively.

At lines 377 to 379, the authors included the description of the studied population in the Results section of their draft. I suggest moving this information to the Participants section (at line 217), as it represents a simple description of the studied population rather than a result aligned with the study's objective.

Experimental design

This current version permits a better understanding of the study methodology. I have few questions:

a) You mentioned that the quality of the surface EMG signal can be influenced by various factors, including subcutaneous fat thickness. Additionally, you quantified BMI and reported that one-third of the studied population was classified as overweight or obese. In this context, I am curious why you did not establish a BMI threshold as an inclusion criterion. Furthermore, it is well known that BMI has certain limitations, particularly regarding the distribution of muscle and fat mass throughout the body. Would it be feasible to use a skinfold caliper to determine a threshold for epidermal thickness in the area of interest?

b) The authors mentioned in line 245 of their manuscript that they assessed heart rate during walking trials in order to exclude participants who exhibited a heart rate higher than the "safe maximum heart rate." Therefore, I would like to know:
- How did you determine the maximum heart rate for each participant? Was it an estimated maximum heart rate (for instance, using the Karvonen equation), or was it determined through a stress test?
- I understand that the authors aimed to assess heart rate during exercise for safety reasons, but this raises further questions regarding the study population and their level of physical activity. Although you worked with physiotherapy students, this information (in my opinion) does not clearly define whether your population is sedentary, athletic, etc. This distinction is crucial, as it can significantly impact the voluntary activation level of your participants and, consequently, directly influence your results.

c) The authors did not specify in their previous response whether participants underwent a habituation phase on the treadmill before performing the experimental conditions. If participants are not accustomed to walking on a treadmill, this could alter both their gait pattern and their muscle activation pattern.

d) Why didn’t the authors include an additional condition to assess the activity of the multifidus and erector spinae muscles at a self-selected speed?

Validity of the findings

As previously mentioned, the results may be influenced by various factors, such as :
- the level of subcutaneous fat, which could affect the EMG data,
- the physical activity level of the studied population,
- or whether participants underwent a sufficiently long habituation phase to stabilize their gait pattern on the treadmill before starting the recordings.

·

Basic reporting

No comment

Experimental design

No comment

Validity of the findings

No comment

Additional comments

No comment

---

## Round 0.3 · Minor Revisions

Upon further review the manuscript has been improved, however, there are further clarifications and amendments needed.

·

Basic reporting

Thank you for your response. Most of the concerns have been resolved, but please address the notation of the hypothesis.

I have reviewed the Tracked Changes (Bryndal_et_al_manuscript_v2.docx), but it appears that the changes noted by the authors do not correspond with the tracked changes. For example, the authors responded that revisions related to prior research were made in lines 124-129, but no such changes were found in those sections. This time, I have inferred the changes based on the authors' responses. To improve readability, please ensure that the line numbers are carefully checked before responding.
* * *
Author Previous Response:
Answer for “New Comments for Point 3”: We appreciate the reviewer’s observation and agree that explicitly stating the hypothesis will further clarify the focus of our study. We have now added a clearly stated hypothesis to the manuscript to better align the study's objectives and conclusions. The revised Introduction section reads as follows (line 143-146):
"Hypothesis: We hypothesize that the activity of motor units in the erector spinae and multifidus muscles increases proportionally with gait speed, and that the growth factor of this activity will demonstrate a decreasing trend at higher speeds, reflecting adaptations in muscle coordination and recruitment patterns."
This addition is included in the Introduction section to provide a clear foundation for the study's objectives.

New Comment
• As mentioned in the first review, the authors did not measure motor unit activity in this study. Therefore, including the term "motor units" in the hypothesis may cause confusion. Please revise the wording in the Introduction, as well as in the Abstract and Discussion (Line 420).
• The term "growth factor" is introduced abruptly in the hypothesis, which may confuse readers. Please rephrase it for better clarity.

Experimental design

An acceptable response has been provided.

Validity of the findings

An acceptable response has been provided.

Additional comments

An acceptable response has been provided.

---

## Round 0.4 · accepted · Accept

An acceptable response has been provided to all reviewers comments. The article is ready to be published. Thanks.